# Irradiation as a Promising Technology to Improve Bacteriological and Physicochemical Quality of Fish

**DOI:** 10.3390/microorganisms11051105

**Published:** 2023-04-23

**Authors:** Eman F. E. Mohamed, Abd El-Salam E. Hafez, Hanan G. Seadawy, Mohamed F. M. Elrefai, Karima Abdallah, Rasha M. El Bayomi, Abdallah Tageldein Mansour, Mahmoud M. Bendary, Abdullah M. Izmirly, Bandar K. Baothman, Khairiah Mubarak Alwutayd, Abdallah F. A. Mahmoud

**Affiliations:** 1Department of Food Hygiene, Safety and Technology, Faculty of Veterinary Medicine, Zagazig University, Zagazig 44519, Egypt; emanfarag995@gmail.com (E.F.E.M.); aahafez@vet.zu.edu.eg (A.E.-S.E.H.); karimaeissa_1989@yahoo.com (K.A.); rmazab@vet.zu.edu.eg (R.M.E.B.); 2Agriculture Research Center (ARC), Animal Health Research Institute (AHRI), Dokki, Giza 3751254, Egypt; hananseadawy70@gmail.com; 3Department of Anatomy, Histology, Physiology and Biochemistry, Faculty of Medicine, Hashemite University, Zarqa 13110, Jordan; mohamedfathianatomy@yahoo.com; 4Department of Anatomy and Embryology, Faculty of Medicine, Ain Shams University, Cairo 11566, Egypt; 5Animal and Fish Production Department, College of Agricultural and Food Sciences, King Faisal University, P.O. Box 420, Al-Ahsa 31982, Saudi Arabia; amansour@kfu.edu.sa; 6Fish and Animal Production Department, Faculty of Agriculture (Saba Basha), Alexandria University, Alexandria 21531, Egypt; 7Department of Microbiology and Immunology, Faculty of Pharmacy, Port Said University, Port Said 42511, Egypt; 8Department of Medical Laboratory Science, Faculty of Applied Medical Science, King Abdulaziz University, P.O. Box 80216, Jeddah 21589, Saudi Arabia; amizmirli@kau.edu.sa; 9Special Infectious Agents Unit-BSL3, King Fahd Medical Research Center, King Abdul-Aziz University, Jeddah 21589, Saudi Arabia; 10Department of Medical Laboratory Technology, Faculty of Applied Medical Sciences in Rabigh, King Abdulaziz University, Jeddah 21589, Saudi Arabia; bbaothman@kau.edu.sa; 11Department of Biology, College of Science, Princess Nourah bint Abdulrahman University, P.O. Box 84428, Riyadh 11671, Saudi Arabia; kmalwateed@pnu.edu.sa

**Keywords:** gamma irradiation, pathogenic bacteria, fish quality, proximate composition

## Abstract

Fish is an excellent source of protein and other essential minerals and vitamins; nevertheless, several food-borne disease outbreaks have been linked to the consumption of different types of fish. Therefore, we aimed to overcome these health threats by evaluating gamma radiation as a good fish preservation method. The aerobic plate count (APC), identification of most common pathogenic bacteria, organoleptic properties, proximate composition, and other chemical evaluations were detected in both untreated and gamma-treated fish. The overall grades of organoleptic evaluations ranged from good to very good. Fortunately, the overall chemical analysis of all examined fish samples was accepted. For the untreated fish samples, the APC was within and above the permissible limit (5 × 10^7^ CFU/g). Pathogenic bacteria were detected with a high prevalence rate, especially *S. aureus*, which was found in high percentages among examined untreated fish samples. Regarding the treated fish samples, APC and pathogenic bacterial counts were reduced in a dose-dependent manner, and the irradiation at dose 5 KGy resulted in complete eradication of the aerobic plate count (not detectable) with a mean reduction percentage equal to 100%. However, gamma irradiation has no significant effect on proximate composition; particularly, carbohydrates, proteins, and lipids were not significantly affected by low and medium doses of radiation. Therefore, gamma irradiation is a highly effective fish preservation method without any effect on fish quality. Additionally, gamma irradiation as a cold process is an attractive technology for solving the problem arising from fish-borne pathogens, and it has been purposed in this study as a cheap and safe method for reducing microbial contamination of fish.

## 1. Introduction

Food-borne illness outbreaks represent a major concern for public health [1,2]. Thus, the detection of microbial pathogens in food is the solution for the prevention and recognition of complications related to health and safety [3]. Several worldwide crises have been recorded owing to the wide spread of food-borne pathogens [4]. One of the most common food categories involved in food-borne illness outbreaks is fish meat [5]. Fish is extensively consumed by humans in several parts of the world. Microbial spoilage, tissue decomposition, and rancidity are the most common hazards correlated with fish meat. Several bacteria, viruses, fungi, parasites, and zoonotic infections can be transmitted via fish and fish products. *Streptococcaceae*, *Mycobacteriaceae*, *Erysipelothricaceae,* and *Staphylococcaceae* families are the common Gram-positive fish-borne infections; meanwhile, several Gram-negative bacteria, including *Vibrionaceae, Pseudomondaceae, Aeromonadaceae, Hafniaceae,* and *Enterobacteriaceae* families are fish-derived zoonotic pathogens [6]. Numerous pathogens, including *Salmonella* sp., *Escherichia coli* (*E. coli*), and *Staphylococcus. aureus (S. aureus)* show high multi-antibiotics resistance indices (MAR) [7,8] and have been detected repeatedly in different kinds of fish [9,10]. Moreover, many outbreaks of food-borne diseases were correlated with fish-borne pathogens. Life-threatening diseases such as salmonellosis, listeriosis, and vibriosis have been attributed to fish and other food products in various studies [11,12,13,14].

Regarding the source of contamination, fish are vulnerable to contamination from different sources, including in their aquatic environment, from sewage pollution in harvesting areas, and/or after being harvested by workers, utensils, and equipment used during transportation, distribution, and food preparation [15]. Several factors can affect outbreaks related to fish-borne infections, such as the type of pathogens (fungi, bacteria, parasites, and viruses), the health status and defensive power of the host, and other environmental factors, such as the degree of water contamination and sewage pollution [16]. Therefore, it is necessary to ensure hygiene, safe handling, and fish safety for the consumer’s health in order to reduce the risk of fish-borne illness outbreaks.

Fish preservation is essential to maintaining fish freshness for a long time and preventing bacterial, enzymatic, and chemical decomposition. Synthetic preservatives were replaced by preservatives from natural sources due to the unpleasant taste, health hazards, and other chemical interactions related to synthetic preservatives. Despite the strong antimicrobial activities of herbal extracts and their diverse mechanisms of action [17] to hinder microbial resistance, they have several drawbacks, such as the relatively high concentration required to be effective, immunological reactions, and variation in consistency and potency. Therefore, both natural and synthetic preservatives should be avoided in food and food products [18]. Regarding the preservation of fish, several traditional methods have been used to extend its shelf-life, such as smoking, marinating, fermentation, salting, and thermal treatments, including freezing, boiling, chilling, refrigeration, drying, steaming, etc. These conventional methods are correlated with undesirable changes, including reduced sensory, nutritional value, and organoleptic properties. Therefore, other alternative methods must be introduced to extend the shelf-life without compromising the organoleptic characteristics of fish [19]. 

Irradiation is a ‘cool’ technique (called cool pasteurization) that does not raise the temperature. Fish that have been irradiated maintain their flavors and aromas. It also avoids the need for chemical treatments such as fumigation or insecticides to manage bacteria and other pests [20]. Irradiation can provide consumers with good quality fish and fish products. Gamma irradiation is a well-known and commonly used fish preservation method that does not cause any increase in temperature, decreases the microbial population, and extends fish shelf life [21]. This process exposes fish to a carefully controlled amount of energy in the form of high-speed particles or rays that reduce the risk of food poisoning, control fish spoilage, and extend the shelf life of fish without any health risk and with minimal effect on nutritional or sensory qualities. This technique has no effect on food taste, color, and odor, and no radioactive residues are left [22]. Therefore, we found it timely and important to use an alternative method rather than the traditional methods of preserving fish. Owing to the above-mentioned crises correlated with poor fish hygiene, we apply a new and safe method of fish preservation (irradiation at different doses) that can reduce microbial loads and eliminate resistant pathogens contaminating the fish meat while keeping their sensory and nutritional properties intact.

## 2. Materials and Methods

### 2.1. Sample Collection

First, 100 muscle portion samples (50 g) of fresh Tilapia nilotica, Mugil cephalus, mackerel, and sardine (25 of each) were randomly collected from different fish markets in Sharkia Province, Egypt. Each sample was separately packaged and kept in sterile plastic Ziplock bags and immediately transported to the Food Hygiene, Safety, and Technology Laboratory within 1 h. Each sample was stomached with 50 mL of 0.1% sterile buffered peptone water and left for 15 min at room temperature. Then, it was suitable for further microbiological evaluation. 

### 2.2. Bacteriological Examination 

Ten-fold serial dilutions were performed for 1 ml from each sample, then carefully added into sterile Petri dishes and mixed with melted standard plate count agar (45 °C). After that, each plate was incubated at 35 °C for 48 h. The aerobic plate count (APC) per gram was calculated on plates containing 30–300 colonies in triplicate (replicated 3 times), and each count was recorded separately [23]. Phenotypic identification of *S. aureus* isolates was based on standard bacteriological methods, including the hemolytic activity on sheep blood agar, DNase activity, coagulase activity, pigmentation and fermentation characters on nutrient agar and mannitol salt agar, respectively, in addition to the microscopic appearance of Gram-positive grape-like coccoid [24] and API 20S identification kit (BioMerieux, Marcy l’Etoile, France). The suspected *E. coli* and *Salmonella* sp. colonies were subjected to phenotypic identification such as microscopic appearance, fermentation activity on MacConkey agar and EMB agar, and Multiple-Tube (MPN) Fermentation Techniques, and cytochrome oxidase, triple sugar iron agar, urea, and indole tests according to the previous studies [25,26,27], respectively. Additionally, the genotypic characterization of these isolates was confirmed by the genetic detection of the specific 16S rRNA genes. The primer sequences for *S. aureus* were [forward: CCTATAAGACTGGGATAACTTCGGG] and [reverse: CTTTGAGTTTCAACCTTGCGGTCG] [28]; meanwhile, CCTACGGGAGGCAGCAG and CCGTCAATTCCTTTRAGTTT were used as forward and reverse primers for the amplification of the salmonella 16S rRNA gene [29]. In the same context, the amplification of the *E. coli* 16S rRNA gene was performed using the following forward and reverse primers: GAAGCTTGCTTCTTTGCT and GAGCCCGGGGATTTCACAT [30]. 

#### 2.2.1. Staphylococcal Enterotoxins Serotyping 

The clear culture supernatant fluid from each *S. aureus* isolate was tested serologically using the Reverse Passive Latex Agglutination technique “RPLA” using kits for the detection of staphylococcal enterotoxins A, B, C, and D (SET-RPLA, Denka Sekeu LTD, Japan), which is based on the detection of soluble antigens through the agglutination of antibody-coated cells or particles [31]. 

#### 2.2.2. Serological Identification of *E. coli*

The isolates were serologically identified according to [26] using rapid diagnostic *E. coli* antisera sets (DENKA SEIKEN Co., Tokyo, Japan) for diagnosis of the enteropathogenic types.

#### 2.2.3. Serological Identification of *Salmonella* sp.

Serological identification of Salmonellae was carried out according to the Kauffman–White scheme [32] for the determination of somatic (O) and flagellar (H) antigens using *Salmonella antiserum* (DENKA SEIKEN Co., Japan). 

### 2.3. Study Design 

Five groups of fish fillet samples were subjected to treatment with gamma irradiation: Group 1: For evaluation of the effectiveness of gamma irradiation on APC of the examined samples; Group 2: For evaluation of the effectiveness of gamma irradiation on the fish fillet samples contaminated with *E. coli*; Group 3: For evaluation of the effectiveness of gamma irradiation on the fish fillet samples contaminated with *Salmonella Typhimurium*; Group 4: For evaluation of the effectiveness of gamma irradiation on the fish fillet samples contaminated with *S. aureus*; Group 5: For evaluation of the effectiveness of gamma irradiation on organoleptic properties, proximate composition, and other physicochemical analysis (pH–TMA–TVBN–TBA) of the examined samples. Each group was sub-divided into four divisions (control untreated division, gamma irradiation treated division at 1 KGy, gamma irradiation treated division at 3 KGy, gamma irradiation treated division at 5 KGy). The procedure was performed in triplicate (replicated 3 times).

### 2.4. Gamma Irradiation Treatment

The irradiation process was carried out at the National Center for Radiation Research and Technology (NCRRT) in Nasr City, Cairo, Egypt. The irradiation facility used was Indian Gamma Cell, and the dose rate was 0.7881 KGy/h. The radiation source was Cobalt^60^, which assured uniform gamma irradiation of the experimental samples. The fish fillet samples were prepared and divided into five groups to be subjected to treatment by gamma irradiation, as mentioned before in the preparation of fish fillets subjected to treatment by gamma irradiation. The tests were replicated 3 times for APC, *E. coli*, *Salmonella Typhimurium*, *S. aureus,* and other analyses (organoleptic evaluation, proximate composition, and other chemical analyses). Upon completing the desired passes, each package was returned to the cooler along with ice and transported to the microbiology laboratory for analysis.

### 2.5. Organoleptic, Proximate Composition, and Other Physicochemical Evaluation for Treated and Untreated Samples 

The organoleptic evaluations were determined using the scoring test improved by [33], where the overall acceptability of fish fillets = 20 (5 for each for appearance, odor, texture, and flavor). It is Excellent (Acceptable), Very Good (Acceptable), and Good (Acceptable) if the score is equal to 20, 18.2–19.9, and 15.2–18.1, respectively. On the other hand, it is Middle, Poor (Borderline), and Spoiled (Unacceptable) if the score is equal to 11.2–15.1, 7.2–11.2, and 4–7.1, respectively. The proximate composition of the studied fish was applied for the determination of moisture, protein, fat, carbohydrates, and ash according to the standard method recommended by the Association of Official Analytical Chemists “AOAC” [34]. The determination of physicochemical properties, including pH, Total Volatile Nitrogen (TVN), Trimethylamine (TMA), Thiobarbituric Acid “TBA”, and histamine level using ELISA was estimated according to [35,36,37,38,39]. 

### 2.6. Bacteriological Analysis of the Treated Samples

After the irradiation, samples were analyzed for bacterial populations. To perform the microbial analysis, first, the sample was blended for 2 min, and then samples were serially diluted in BPW 0.1%. The dilutions were plated in duplicate onto count plates and then incubated at 37 °C for 24 h. Finally, APC, *E. coli*, *Salmonella Typhimurium*, and *S. aureus* were counted, and the count was expressed as log10 CFU/g. 

### 2.7. Determination of Reductions 

The log reductions in the experiments were calculated using the following formula: log reduction = log10 initial concentration–log10 final concentration. The reductions in single experiments on fish fillets for APC, *E. coli*, *Salmonella Typhimurium*, *S. aureus* count were calculated based on the average colony forming units per milliliter before and after the gamma irradiation treatment, and the final reductions were based on the average reductions from the test replications. Reduction% = (Mean of control − Mean of treated sample) 100/ Mean of control.

### 2.8. Statistical Analysis

The values were presented as means ± standard error (SE). The data were subjected to the statistical package for social sciences (SPSS-16.; Chicago, IL, USA) software and One-way Analysis of Variance (ANOVA) at a 95% level of confidence. Significant differences among the means were determined by Tukey’s Kramer HD test considering *p* < 0.05 as significant.

## 3. Results

### 3.1. Bacteriological Examination of Untreated and Treated Fish Samples 

According to the aerobic plate count (APC), the sardine fish fillets, in contrast to mullet, showed the highest APC. The mean APC count for all tested samples was (5.21 × 10^8^ CFU/g). Unfortunately, all samples (100) in this study showed growth on an aerobic agar plate. Based on the standard bacteriological methods, 3 pathogens (*S. aureus*, *E. coli*, and *Salmonella* sp.) were found among the tested fish fillets samples, and they were confirmed by genetic detection of the 16S rRNA gene, as shown in Table 1. The tilapia and mullet fish fillets were the most common samples contaminated with *S. aureus*, *E. coli*, and *Salmonella* sp., as shown in Table 1 and Figure 1. Surprisingly, all our samples were contaminated with at least one of the tested pathogens. The mean viable counts detected for all tested pathogens, *S. aureus* and *E. coli*, in contrast to *Salmonella* sp. (6 × 10^5^ CFU/g), were detected with relatively higher values (6 × 10^6^ CFU/g). Regarding the serological analysis for the tested pathogens, only the staphylococcal enterotoxin A, C, and D, in addition to Enterohemorrhagic *E. coli* (EHEC) and Enteropathogenic *E. coli* (EPEC) strains, were detected in all tested fish fillets. In the same context, only *Salmonella enteritidis* was identified for all detected *Salmonella* sp. 

Regarding the irradiated fish samples, as expected, the APC was reduced upon exposure to gamma irradiation in a dose-dependent manner. At least more than one log cycle (90%) was reduced for the lowest dose of radiation (1 KGy). The reduction percentages were 99.08% and 99.99% for doses 1 and 3 KGy. Regarding *S. aureus* count, the mean initial count of *S. aureus* in fish fillets was 6 × 10^6^ CFU/g. Interestingly, the reduction percentages for doses 1 and 3 KGy were 90.55% and 99.17%, respectively. For *Salmonella Typhimurium* and *E. coli* count, the reduction percentages for doses 1 and 3 KGy were (90.71%, 99.92%) and (90.80%, 99.93%), respectively, as shown in Table 2. Therefore, irradiation at doses 1 and 3 KGy had a significant effect (*p* < 0.05) on APC and all other tested pathogens. It is great that all aerobic bacteria and other tested pathogens were killed when exposed to 5 KGy gamma irradiation with a mean reduction percentage equal to 100%, as shown in Figure 1 and Table 2.

### 3.2. Organoleptic Evaluation for Untreated and Treated Fish Samples

The organoleptic properties for all tested untreated fish fillets were acceptable, as Poor and Spoiled grades were not observed among all samples. Very Good and Good color, texture, and flavor were recorded for all tested samples, as shown in Table 3.

For the treated fish samples, the gamma irradiation did not impair the organoleptic properties of the treated fish fillet samples. Poor and Spoiled grades were not observed; meanwhile, Very Good and Good color, texture, and flavor were recorded for all groups treated with dose 1 KGy. On the other hand, the group treated with 3 KGy and 5 KGy showed Good and Middle organoleptic properties, respectively, as shown in Table 4. 

### 3.3. Physicochemical Properties of Untreated and Treated Fish Samples 

Regarding the physicochemical properties of the tested fish samples, as shown in Table 5, the pH, Total Volatile Nitrogen “TVN”, Trimethylamine ™, and Thiobarbituric Acid Number “TBA”, in addition to histamine levels were determined. Among the tested physicochemical parameters, Total Volatile Nitrogen (TVN) was detected with the highest values for all types of fish fillet samples.

Considering the physicochemical properties of the irradiated fish samples, all the chemical measures of the examined fish samples were reduced in a dose-dependent manner in all tested tilapia, mullet, mackerel, and sardine, as shown in Table 6 and Figure 2.

### 3.4. The Proximate Composition for Treated and Untreated Fish Samples

The proximate composition of the treated and untreated fish samples is shown in Figure 3. The moisture content was increased in a dose-dependent manner among all types of fish. In contrast to tilapia, sardine has the highest percentage of moisture content. The percentage of fat and protein content is almost constant after being treated with different doses of gamma radiation; therefore, gamma radiation has no effect on the fat and protein content of fish. Irregularly, the contents of carbohydrates and ash were not correlated with the gamma radiation treatment.

## 4. Discussion

Fish-borne infections and toxication is the most important crisis affecting seafood industries. The high nutritional value of seafood makes it an excellent choice for the health-conscious and nutritionists owing to its high-quality protein and omega-3 fatty acids contents. On the other hand, seafood-borne infections and seafood poisoning, as well as retail sheep meat [40], reduces progress in the seafood and processed meat industries. These infections can occur from the source or during processing and packaging. The fish defense system breaks down after death, allowing the invasion and multiplication of several bacteria that exceed the permissible limit (PL) recommended by [41], especially after being harvested, through unhygienic handling, transportation, and marketing, in addition to storage [42]. In addition to the reduction in nutritional values and fish quality, severe health hazards can develop from the presence of pathogens in the fish. Therefore, there is a challenge to prevent the spread of fish-borne infections [43,44] and to provide the community with safe and acceptable fish. Therefore, proper fish preservation techniques must be developed and applied. To address the illustrated gap between the benefits of seafood and their health crises, we tried to break through this issue by evaluating the effectiveness of gamma irradiation on fish quality and investigating the effects of different doses on existing food-borne pathogens. Several previous studies tested the effectiveness of gamma radiation as a fish preservation method [45,46]. However, the novelty of our study was in evaluating the effectiveness of different irradiation doses, not limited only to the total microbial counts or their effect on the most prominent fish-borne pathogens, but several other parameters such as proximate composition, organoleptic, and other physicochemical properties in order to select the suitable radiation dose with higher preservation capacity and showing the acceptable proximate composition, organoleptic, and other physicochemical properties of irradiated fish.

Aerobic plate count (APC) is considered an indicator of the overall degree of microbial contamination in food [47]. APC acceptability, according to the permissible limit recommended by [41], is not more than 5 × 10^7^ CFU/g for chilled and frozen fish. The results illustrated in this study for the untreated fish were within and above the permissible limit, at nearly the same range as previous reports [48,49], and higher than other studies [50,51]. The high count of *S. aureus* and high percentages of isolation among different examined fish samples in this and other studies [52,53] were due to fish contamination during harvesting and subsequent unhygienic practices during handling and processing, as *S. aureus* does not normally appear as a part of the natural microflora of newly caught marine and cultivated fish. On the other hand, a higher prevalence rate of *E. coli* and *Salmonella* sp. was recorded in contrast to previous studies [54,55,56,57]. The difference in the microbiological contamination levels of fish may be attributed to variations in handling practices and the possible risk of contamination from water sources.

Several basic fish decontamination and preservation methods were applied, such as heating, freezing, controlling water activity, and irradiating. In this research, we introduced evidence for the priority of gamma irradiation over other preservation methods. Concerning the microbiological quality of fish, this particular point gives utmost importance to the use of gamma rays as an excellent fish preservation method. Although freezing technology is cheaper and more convenient for fish preservation, the microbiological quality of the preserved fish can prevent the wide application of this method. This was confirmed by previous studies which detected coliform bacteria with higher CFU/g than the permitted level [50,58]. One of the oldest fish preservation processes is controlling water activity through salting technology. Several authors announced that 60% of the examined salted fish samples had a high APC count of 10^5^:10^6^ [59,60]. The poor microbiological quality of the examined smoked fish, which contained fungal toxins, was documented in a previous report [61]. Considerable coliform count with a low CFU/g level was also recorded among fish samples preserved by heating [61]. On the other hand, several reports evaluated the efficiency of gamma radiation (1, 3, 5, and 8 KGy) on fish and fish products and confirmed that the APC was affected by this radiation [62,63]. Interestingly, in this study, the APC and the pathogenic bacterial count were reduced in a dose-depending manner, and the irradiation at dose 5 KGy caused complete eradication of aerobic plate count (not detected) with a mean reduction percentage equal to 100%. The highest and lowest effects of gamma irradiation were recorded in the contaminated fish samples with *E. coli* and *S. aureus,* respectively.

Regarding the organoleptic properties of irradiated fish, the gamma irradiation did not impair the organoleptic properties of the treated fish fillet samples. Poor and Spoiled grades were not observed among the tested fish fillets. Generally, the organoleptic scores of the fish samples treated with gamma radiation slowly improved in a dose-dependent manner. In a previous report, fish samples irradiated with a medium dose (4–5 KGy) maintained their sensory properties and extended their acceptability compared to non-irradiated fish [64]. Other preservation methods reduce fish organoleptic scores. For example, heating technology causes the texture to become very tough and causes dehydration of fish muscle and shrinkage of fish filaments [65]. Smoking is a preservation method that provides heat and antimicrobial smoke chemicals to reduce the water activity in fish. This preservation method can improve fish organoleptic properties; however, the antimicrobial chemicals can react with essential nutrients in the fish, leading to a reduction in nutritional values [66]. Moreover, some carcinogenic properties of these chemicals and environmental pollution have significant impacts on the wide application of this preservation method. In the same context, preservation methods depending on freezing technology can maintain the organoleptic properties of fish but cannot keep good microbiological quality, as psychrotrophic pathogens can grow [67].

Considering the effect of other preservation methods on the proximate composition, heat preservation can alter lipid composition and decrease omega-3 fatty acids, increase the denaturation of proteins, and cause significant loss of minerals, essential amino acids, and vitamins [68]. Controlling water activities by drying has a great negative effect on proximate composition, especially protein content, and can affect the valuable nutritional content of fish. Additionally, fish lipid oxidation is a common phenomenon caused by drying and salting [62]. Alteration of the physical and chemical properties of protein is common in smoked fish. Loss of lysine amino acid is well documented due to the smoking process [69]. Of note, gamma irradiation has no significant effect on the proximate composition. The carbohydrates, proteins, and lipids are not significantly affected by low- and medium-range doses of radiation according to their nutrient content and digestibility [70,71,72]. In accordance with these reports, we found that there is no significant difference (*p* < 0.05) in proximate composition between the irradiated and non-irradiated samples. Additionally, the physicochemical indicator (pH, TVB-N, TMA-N, and TBARS) values of irradiated samples were lower than those of non-irradiated samples but with no statistically significant effect (*p* > 0.05) of irradiation at doses 1 and 3 KGy on the examined fish samples, as recorded in this study and other studies [73,74,75].

Irradiation is a cold process that is well-established for solving the problem arising from the consumption of contaminated foods of animal origin. It has been purposed as a new and safe method for reducing microbial contamination of meat [76,77]. The effects of gamma irradiation on sensory properties vary depending on the type of food being irradiated [78]. Therefore, we recommend using 3 KGy of gamma radiation to reduce the microbial population without any effect on proximate composition, organoleptic, and other physicochemical properties. Additionally, we also suggest applying 5 KGy of gamma radiation to completely eradicate the microbial count with acceptable changes in proximate composition, organoleptic, and other physicochemical properties when fish or fish products are used for immunocompromised individuals or in case of the wide spreading of epidemic and endemic diseases. Finally, gamma irradiation is an attractive developing technology for keeping the fish’s safety, sensory, and nutritional properties. Finally, we found that the future perspective application of irradiation in the preservation of fish and other food products is developing owing to the increasing demand for meats and seafood. Of note, it is necessary to provide consumers with minimally processed fish products with acceptable microbiological quality. The costs of irradiation and the advantages of non-thermal preservation technology, in addition to the drawbacks of heating, freezing, and controlling water activity preservation methods, give priority to irradiation technology to be used worldwide. Moreover, most international trade requirements are met in irradiating seafood, so there are great opportunities for the practical application of gamma irradiation in the seafood exporting industry. The expansion of gamma irradiation applications has gained the attention of many researchers, not only for seafood but for a wide range of fruits and vegetables [79]. Furthermore, it can be used to increase the sanitary–hygienic conditions of production rooms and prevent sprouting in potatoes and in the preservation of ready-to-eat baby spinach leaves [80,81].

## 5. Conclusions

The application of gamma irradiation is a highly effective method for the inactivation, reduction, and eradication of pathogenic and spoilage microorganisms in a dose-dependent manner. Therefore, it can be used as a control measure to lower the health risks of fish-borne pathogens without changing the organoleptic, proximate composition, and physic-chemical properties of fish.

## Figures and Tables

**Figure 1 microorganisms-11-01105-f001:**
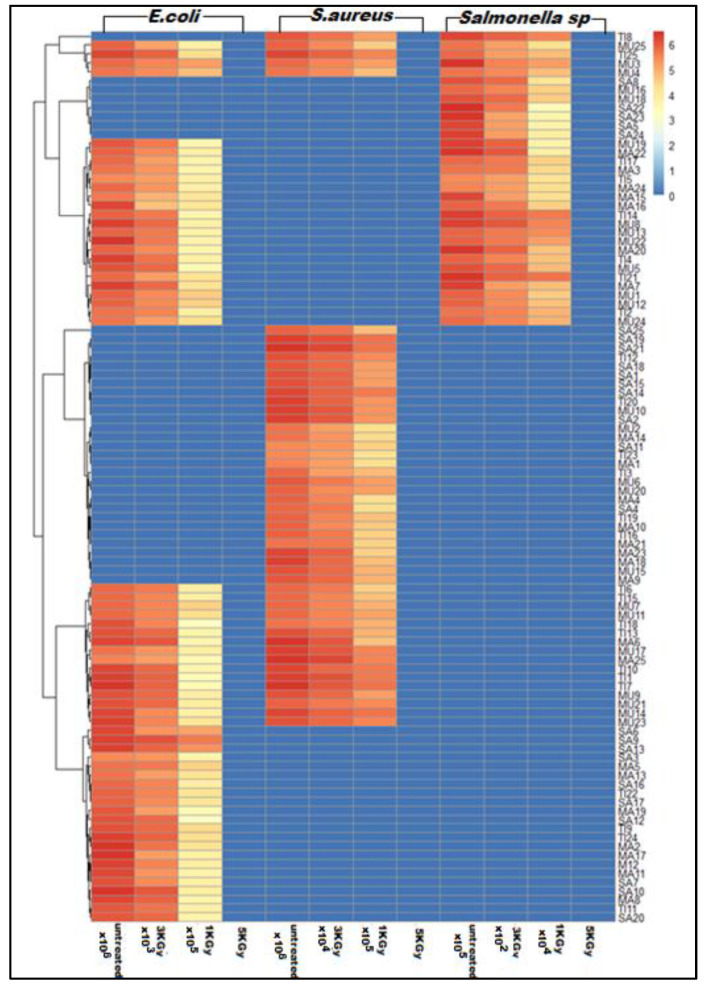
Heat map reflecting the impact of different doses of gamma radiation (1, 3, and 5 KGy) on the viable count of pathogenic bacteria (*E. coli*, *S.aureus*, and *Salmonella enteritidis*). The color key indicates the viable count of pathogenic bacteria among treated and untreated fish fillet samples measured as CFU/g; the color code ranges from darker blue (0 CFU/g) to darker red color (6 CFU/g). The sample codes on the right side of the dendrogram consist of the first 2 letters of each sample (TI: tilapia, MU: mullet, MA: mackerel, SA: sardine), followed by the number according to the order of detection and denote the type of samples, TI: tilapia, MU: mullet, MA: mackerel, SA: sardine.

**Figure 2 microorganisms-11-01105-f002:**
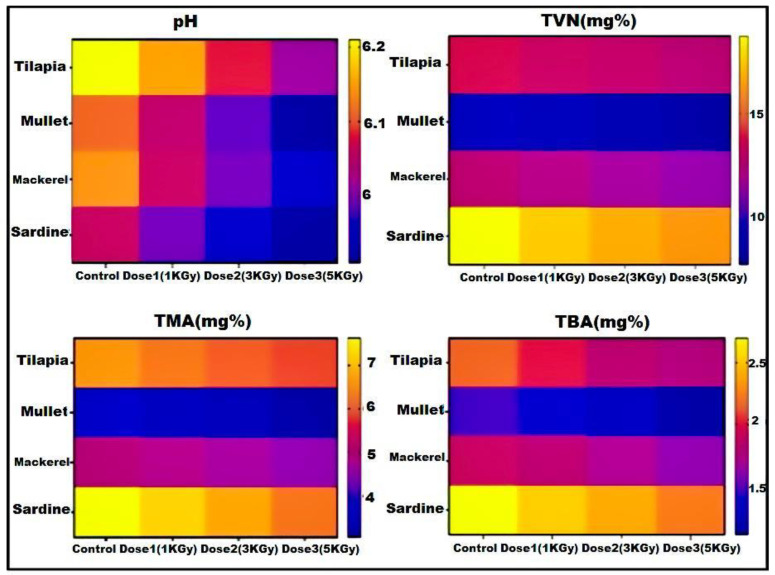
Effect of radiation with different doses on chemical analysis of the examined fish. The color code reflects the value for each parameter regarding the untreated samples (control) and irradiated samples with different doses (1, 3, and 5 KGy). The dark blue color reflects the lowest value; meanwhile, the dark yellow color reflects the highest value.

**Figure 3 microorganisms-11-01105-f003:**
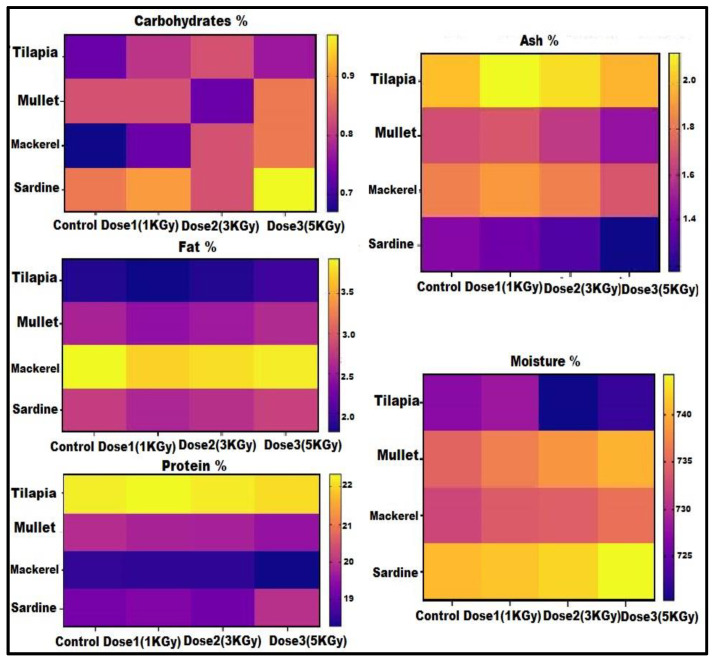
Effect of radiation with different doses on the proximate composition of the examined fish. The values of the proximate composition for the untreated samples (control) and irradiated samples with different doses (1, 3, and 5 KGy) are represented using color code. The dark yellow color reflects the highest value; meanwhile, the dark blue color reflects the lowest value.

**Table 1 microorganisms-11-01105-t001:** The number of untreated fish fillet samples contaminated with *S. aureus, E. coli*, and *Salmonella* sp.

Type of Fish Fillet (NO.)	*S. aureus*	*E. coli*	*Salmonella* sp.
**Tilapia (25)**	**15 ± 0**	**18 ± 0**	**8 ± 0**
**Mullet (25)**	**15 ± 0**	**18 ± 0**	**13 ± 0**
**Mackerel (25)**	**10 ± 0**	**17 ± 0**	**7 ± 0**
**Sardine (25)**	**10 ± 0**	**10 ± 0**	**5 ± 0**
**Total**	**50 ± 0**	**63 ± 0**	**33 ± 0**

**Table 2 microorganisms-11-01105-t002:** Effect of gamma radiation with different doses on the aerobic plate count (APC) and other pathogenic bacteria on the examined fish samples.

Irradiation Dose	Mean ± SE	Reduction Count	Reduction%
**APC**			
**Control**	5.21 ± 0.31 ^a^ (1 × 10^8^)		
**1 KGy**	4.75 ± 0.26 ^a^ (1 × 10^6^)	51.62 × 10^7^	99.08%
**3 KGy**	2.88 ± 0.00 ^b^ (1 × 10^4^)	52.09 × 10^7^	99.99%
**5 Kgy**	ND	5.21 × 10^8^	100%
***S. aureus* count**			
**Control**	6.00 ± 0.24 ^a^ (1 × 10^6^)		
**1 KGy**	5.67 ± 0.26 ^b^ (1 × 10^5^)	5.43 × 10^6^	90.55%
**3 KGy**	4.97 ± 0.44 ^c^ (1 × 10^4^)	5.95 × 10^6^	99.17%
**5 KGy**	ND	6.00 × 10^6^	100%
** *Sal. Typhimurium* **			
**Control**	6.00 ± 0.28 ^a^ (1 × 10^5^)		
**1 KGy**	5.57 ± 0.27 ^b^ (1 × 10^4^)	5.44 × 10^5^	90.71%
**3 KGy**	4.69 ± 0.53 ^c^ (1 × 10^2^)	5.99 × 10^5^	99.92%
**5 Kgy**	ND	6.00 × 10^5^	100%
** *E. coli* **			
**Control**	6.00 ± 0.26 ^a^ (1 × 10^6^)		
**1 KGy**	5.52 ± 0.28 ^b^ (1 × 10^5^)	5.44 × 10^6^	90.80%
**3 KGy**	3.98 ± 0.45 ^c^ (1 × 10^3^)	5.99 × 10^6^	99.93%
**5 KGy**	ND	6.00 × 10^6^	100%

Values are mean ± SE. Means within the same column carrying different superscripts are significantly different at (*p* < 0.05) based on Tukey’s Kramer HD test. S.E. = Standard error of mean, % = Percentage, ND = Not Detected. Reduction count = Mean of control (log value)–Mean of treated sample (log value).

**Table 3 microorganisms-11-01105-t003:** The organoleptic evaluation of the examined untreated fish.

Examined Fish	Color (5)	Odor (5)	Texture(5)	Flavor (5)	Overall (20)	Grade
**Tilapia**	4.8 ± 0.01	4.2 ± 0.05	4.4 ± 0.01	4.6 ± 0.04	18 ± 0.03	Good
**Mullet**	5.0 ± 0.00	4.6 ± 0.01	5.0 ± 0.00	4.8 ± 0.00	19.4 ± 0.01	Very good
**Mackerel**	4.8 ± 0.05	4.6 ± 0.04	4.6 ± 0.03	4.8 ± 0.03	18.8 ± 0.04	Very good

The score for sensory evaluation of fish fillets: Overall acceptability = 20 (5 for each of appearance, odor, texture, and flavor); 20 = Excellent; 18.2–19.9 = Very Good; 15.2–18.1 = Good; 11.2–15.1 = Middle; 7.2–11.2 = Poor (Borderline); Spoiled = 4–7.1.

**Table 4 microorganisms-11-01105-t004:** The organoleptic evaluation of the irradiated fish samples.

Dose	Fish Fillets	Color (5)	Odor (5)	Texture (5)	Flavor (5)	Overall (20)	Grade
**1KGy**	**Tilapia**	4.6 ± 0.02	4.2 ± 0.01	4.4 ± 0.03	4.6 ± 0.04	17.8 ± 0.03	Good
	**Mullet**	4.8 ± 0.00	4.4 ± 0.00	4.8 ± 0.01	4.6 ± 0.00	18.6 ± 0.01	Very good
	**Mackerel**	4.8 ± 0.01	4.4 ± 0.02	4.4 ± 0.05	4.6 ± 0.01	18.2 ± 0.02	Very good
	**Sardine**	4.4 ± 0.05	3.8 ± 0.02	4.2 ± 0.01	4.0 ± 0.03	16.4 ± 0.03	Good
**3 KGy**	**Tilapia**	4.4 ± 0.01	3.8 ± 0.01	4.0 ± 0.04	4.2 ± 0.05	16.4 ± 0.03	Good
	**Mullet**	4.6 ± 0.01	4.4 ± 0.03	4.6 ± 0.02	4.6 ± 0.00	18.2 ± 0.02	Very good
	**Mackerel**	4.6 ± 0.02	4.4 ± 0.02	4.2 ± 0.00	4.4 ± 0.00	17.6 ± 0.02	Good
	**Sardine**	4.0 ± 0.03	3.8 ± 0.05	3.8 ± 0.03	3.6 ± 0.04	15.2 ± 0.04	Good
**5 KGy**	**Tilapia**	4.0 ± 0.03	3.4 ± 0.01	3.6 ± 0.03	4.0 ± 0.03	15.0 ± 0.02	Middle
	**Mullet**	4.4 ± 0.00	4.2 ± 0.04	4.6 ± 0.00	4.4 ± 0.02	17.8 ± 0.03	Good
	**Mackerel**	4.2 ± 0.03	4.0 ± 0.06	4.0 ± 0.04	4.4 ± 0.05	16.6 ± 0.05	Good
	**Sardine**	3.8 ± 0.02	3.4 ± 0.03	3.4 ± 0.00	3.2 ± 0.01	13.8 ± 0.02	Middle

The score for sensory evaluation of fish fillets: Overall acceptability = 20 (5 for each of appearance, odor, texture, and flavor); Excellent (20); Very Good (18.2–19.9); Good (15.2–18.1); Middle (11.2–15.1); Poor (Borderline) (7.2–11.2); Spoiled (4–7.1).

**Table 5 microorganisms-11-01105-t005:** Physicochemical properties of the untreated fish samples.

	Minimum	Maximum	Mean ± SE
**pH**			
**Tilapia**	6.08	6.36	6.21 ± 0.08 ^a^
**Mullet**	5.97	6.22	6.11 ± 0.07 ^a^
**Mackerel**	6.03	6.25	6.14 ± 0.06 ^a^
**Sardine**	6.14	6.49	6.31 ± 0.1 ^a^
**TVN (mg%)**			
**Tilapia**	10.3	17.6	13.5 ± 2.15 ^ab^
**Mullet**	6.3	9.2	8 ± 0.87 ^b^
**Mackerel**	9.8	14.7	12.6 ± 1.46 ^ab^
**Sardine**	17.1	21.5	19 ± 1.31 ^a^
**TMA (mg%)**			
**Tilapia**	6.1	7.00	6.57 ± 0.26 ^a^
**Mullet**	3.2	4.00	3.53 ± 0.24 ^c^
**Mackerel**	4.5	5.7	5.03 ± 0.35 ^b^
**Sardine**	6.8	8.2	7.63 ± 0.43 ^a^
**TBA (mg%)**			
**Tilapia**	1.9	2.5	2.17 ± 0.18 ^ab^
**Mullet**	1.3	1.6	1.43 ± 0.09 ^c^
**Mackerel**	1.7	2.2	1.9 ± 0.15 ^bc^
**Sardine**	2.4	3.1	2.7 ± 0.21 ^a^
**Level of Histamine (Mg/100 g)**			
**Tilapia**	1.89	5.86	4.24 ± 5.18 ^a^
**Mullet**	0.25	0.75	0.35 ± 1.00 ^c^
**Mackerel**	0.25	2.87	1.31 ± 4.31 ^b^
**Sardine**	0.25	8.18	1.84 ± 0.00 ^b^

In each individual criterion: Means within the same column carrying different superscripts are significantly different at (*p* < 0.05) based on Tukey’s Kramer HD test. S.E. = Standard error of mean, % = Percentage, mg = milligram, pH = potential of hydrogen or power of hydrogen, TVN = Total Volatile Nitrogen, TMA = Trimethylamine, TBA = Thiobarbituric acid.

**Table 6 microorganisms-11-01105-t006:** Effect of radiation with different doses on physicochemical properties of the examined samples.

		Tilapia	Mullet	Mackerel	Sardine
**pH**	**Control**	6.21 ± 0.08 ^aA^	6.11 ± 0.07 ^aA^	6.14 ± 0.06 ^aA^	6.31 ± 0.10 ^aA^
	**Dose 1 (1 KGy)**	6.15 ± 0.09 ^aA^	6.05 ± 0.08 ^aA^	6.06 ± 0.06 ^aA^	6.23 ± 0.10 ^aA^
	**Dose 3 (3 KGy)**	6.08 ± 0.09 ^aA^	5.98 ± 0.09 ^aA^	5.99 ± 0.04 ^aA^	6.15 ± 0.11 ^aA^
	**Dose 5 (5 KGy)**	6.01 ± 0.08 ^aA^	5.91 ± 0.07 ^aA^	5.95 ± 0.04 ^aA^	6.07 ± 0.10 ^aA^
**TVN (mg%)**	**Control**	13.50 ± 2.15 ^abA^	8.00 ± 0.87 ^bA^	12.60 ± 1.46 ^abA^	19.00 ± 1.31 ^aA^
	**Dose 1 (1 KGy)**	13.10 ± 2.12 ^abA^	7.67 ± 0.96 ^bA^	11.87 ± 1.38 ^abA^	17.67 ± 1.20 ^aA^
	**Dose 3 (3 KGy)**	12.80 ± 2.14 ^abA^	7.37 ± 1.00 ^bA^	11.33 ± 1.34 ^abA^	16.90 ± 1.15 ^aA^
	**Dose 5 (5 KGy)**	12.50 ± 2.06 ^abA^	7.17 ± 1.05 ^bA^	10.93 ± 1.33 ^abA^	16.27 ± 1.15 ^aA^
**TMA (mg%)**	**Control**	6.57 ± 0.26 ^aA^	3.53 ± 0.24 ^cA^	5.03 ± 0.35 ^bA^	7.63 ± 0.43 ^aA^
	**Dose 1 (1 KGy)**	6.27 ± 0.32 ^aA^	3.33 ± 0.20 ^cA^	4.83 ± 0.34 ^bA^	7.20 ± 0.50 ^aA^
	**Dose 3 (3 KGy)**	6.03 ± 0.29 ^aA^	3.20 ± 0.21 ^cA^	4.67 ± 0.27 ^bA^	6.73 ± 0.43 ^aA^
	**Dose 5 (5 KGy)**	5.87 ± 0.26 ^aA^	3.03 ± 0.19 ^cA^	4.47 ± 0.22 ^bA^	6.23 ± 0.48 ^aA^
**TBA (mg/kg)**	**Control**	2.17 ± 0.18 ^abA^	1.43 ± 0.09 ^cA^	1.90 ± 0.15 ^bcA^	2.70 ± 0.21 ^aA^
	**Dose 1 (1 KGy)**	2.00 ± 0.15 ^abA^	1.37 ± 0.09 ^bA^	1.83 ± 0.19 ^bA^	2.53 ± 0.20 ^aA^
	**Dose 3 (3 KGy)**	1.83 ± 0.13 ^abA^	1.20 ± 0.10 ^bA^	1.70 ± 0.15 ^bA^	2.40 ± 0.21 ^aA^
	**Dose 5 (5 KGy)**	1.77 ± 0.12 ^abA^	1.10 ± 0.06 ^cA^	1.60 ± 0.15 ^bcA^	2.23 ± 0.20 ^aA^
**Histamine** **level**	**Control**	25.00 ± 0.00 ^a^	22.00 ± 0.00 ^a^	24.00 ± 0.00 ^a^	26.00 ± 0.00 ^a^
	**Dose 1 (1 KGy)**	11.90 ± 1.46 ^b^	9.43 ± 1.55 ^b^	10.70 ± 1.76 ^b^	12.60 ± 1.46 ^b^
	**Dose 3 (3 KGy)**	8.13 ± 0.50 ^c^	7.33 ± 0.30 ^c^	9.16 ± 0.52 ^c^	10.22 ± 0.40 ^c^
	**Dose 5 (5 KGy)**	5.27 ± 0.78 ^c^	4.27 ± 0.28 ^c^	7.29 ± 0.71 ^c^	8.45 ± 0.58 ^c^

In each individual criterion: Rows of the same radiation dose among tilapia, mullet, mackerel, and sardine fish carrying different small superscript letters (a, b, c) are significantly different from each other at *p* < 0.05 (test of the effect of fish type). Columns carrying the same large superscript letters (A) are not significantly different at *p* > 0.05 (test of the effect of the treated groups) based on Tukey’s Kramer HD test. S.E. = Standard error of mean, % = Percentage, mg = milligram. pH = potential of hydrogen or power of hydrogen, TVN = Total Volatile Nitrogen, TMA = Trimethylamine, TBA = Thiobarbituric acid.

## Data Availability

All obtained data in this work are included in the submitted manuscript.

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
