# Peer review of "Irradiation as a Promising Technology to Improve Bacteriological and Physicochemical Quality of Fish"

_microorganisms, 2023, doi:10.3390/microorganisms11051105_

Round 1
Reviewer 1 Report (Previous Reviewer 1)
Even though during the revised version, the authors found an unintended error in the reduction count and reduction percentage calculation, in my opinion, the error are still existing.
I am really sorry, but in table 2 (from actual version, line 247) the reduction count and the reduction % is wrong calculated.
First it is not correct to write 51.62x107 and 52.09 x107 (in the APC, 1kGy and 3 kGy, respectively).
Secondly, if you have in the control 5.21 x 108 UFC/g (the total bacterial population found before the irradiation) and after 1kGy, the total bacterial population is 4.75 x 106 UFC/g, the reduction % is impossible to be 99.08%.
The same error occurs in all the calculation presented in table 2.
Author Response
Dear editor, in the line of the previous decision of this manuscript “ minor revision” we added more information in order to meet the expectation of the reviewers. All the reviewer and academic editor comment were taken in our consideration in this new version of this manuscript. The following are the response to the reviewer comments in details.
Reviewer 1
Even though during the revised version, the authors found an unintended error in the reduction count and reduction percentage calculation, in my opinion, the error are still existing.
I am really sorry, but in table 2 (from actual version, line 247) the reduction count and the reduction % is wrong calculated. First it is not correct to write 51.62x107 and 52.09 x107 (in the APC, 1kGy and 3 kGy, respectively).Secondly, if you have in the control 5.21 x 108 UFC/g (the total bacterial population found before the irradiation) and after 1kGy, the total bacterial population is 4.75 x 106 UFC/g, the reduction % is impossible to be 99.08%. The same error occurs in all the calculation presented in table 2.
Response to the reviewer comment: Dear reviewer: thank you very much for your comment we carefully revised the calculation of the table 2 several times and we did not find any error in this calculation. The equation is
Reduction % = (Mean of control – Mean of treated sample) 100/ Mean of control.
The mean of control is 5.21x108
The mean of treatment is (5.21x108 - 4.75x106) 100/5.21x 108 = 99.08%
Generally, one log reduction is equal to 90% reduction
Therefore, by using the calculator all our calculations were correct
Reviewer 2 Report (Previous Reviewer 2)
My comments have been addressed and this manuscript can thus be accepted.
Author Response
Reviewer 2
My comments have been addressed and this manuscript can thus be accepted.
Response to the reviewer comment: Thank you very much
Reviewer 3 Report (New Reviewer)
Totally, this article is comprehensive and reliable, but the details need to be revised and improved.
In the abstract, the purpose, method, results and conclusions of this paper are introduced in detail.
The logic of the introduction is reasonable, the research has market demand, and the purpose is clear.
The source and sampling position of the experimental animals are very clear, the index determination method is correct, and the data processing is reliable.
In this paper, the data is expressed clearly, the previous research results are analyzed reasonably, and the conclusion is credible. But some details need to be modified.
Author Response
Reviewer 3
Totally, this article is comprehensive and reliable, but the details need to be revised and improved.
In the abstract, the purpose, method, results and conclusions of this paper are introduced in detail.
Response to the reviewer comment: thank you for your comment we reduced the detail information in this section in accordance to your point of view
The logic of the introduction is reasonable, the research has market demand, and the purpose is clear.
Response to the reviewer comment: thank you for your comment
The source and sampling position of the experimental animals are very clear, the index determination method is correct, and the data processing is reliable.
Response to the reviewer comment: thank you for your comment
In this paper, the data is expressed clearly, the previous research results are analyzed reasonably, and the conclusion is credible. But some details need to be modified.
Response to the reviewer comment: thank you for your comment
Round 2
Reviewer 1 Report (Previous Reviewer 1)
Good luck!
This manuscript is a resubmission of an earlier submission. The following is a list of the peer review reports and author responses from that submission.
Round 1
Reviewer 1 Report
Even though the manuscript was revised by the authors, there still are many confusion details regarding the following:
- The identification of pathogenic bacteria using “standard bacteriological method “ (line 134 and 230).
- The way of how results were expressed
- The results are expressed wrong in CFU/mL (“6×106 CFU/ml.”-line 288)
I do not understand why the results from the present manuscript are different from those presented on the first version? (lines 285-291; Table 3).
The samples code from figure 1 are not understandable.
Not all the figures have legend.
Reviewer 2 Report
This manuscript focused on the application of irradiation in improving the biological and physicochemical quality of fish. The methods were presented in detail and the organization of this manuscript was reasonable. It was observed that irradiation was effective to inactivate bacterial pathogens and to improve fish quality. Therefore, I would suggest a revision of this manuscript. Detailed comments are as below:
(1) Lines 2-3: Please consider changing the title to “Irradiation as a promising technology to improve bacteriological and physicochemical quality of fish”.
(2) Line 34: What is the permissible limit?
(3) There are many typos in this manuscript. For example, gram+ve in line 58 and E.coli in line 131. Please check it throughout this manuscript.
(4) Table 1: How many replicates were conducted for this test? The means ± standard error should be presented.
(5) Results: The bacteriological, organoleptic, and physicochemical properties of untreated fish samples served as the control. Thus, the related data should be combined with that in irradiation-treated groups.
(6) Line 469: Could you please give some future perspectives regarding the application of irradiation in fish preservation?